# The Key Role of Tin (Sn) in Microstructure and Mechanical Properties of Ti_2_SnC (M_2_AX) Thin Nanocrystalline Films and Powdered Polycrystalline Samples

**DOI:** 10.3390/nano12030307

**Published:** 2022-01-18

**Authors:** Snejana Bakardjieva, Jiří Plocek, Bauyrzhan Ismagulov, Jaroslav Kupčík, Jiří Vacík, Giovanni Ceccio, Vasily Lavrentiev, Jiří Němeček, Štefan Michna, Robert Klie

**Affiliations:** 1Institute of Inorganic Chemistry of the Czech Academy of Sciences, 250 68 Husinec-Rez, Czech Republic; plocek@iic.cas.cz (J.P.); ismagulov@iic.cas.cz (B.I.); kupcik@iic.cas.cz (J.K.); 2Faculty of Mechanical Engineering, JE Purkyně University, Pasteurova 1, 400 96 Ústí nad Labem, Czech Republic; stefan.michna@ujep.cz; 3Department of Inorganic Chemistry, Faculty of Science, Charles University in Prague, Albertov 6, 128 43 Prague, Czech Republic; 4Nuclear Physics Institute, Czech Academy of Sciences, 250 68 Husinec-Rez, Czech Republic; vacik@ujf.cas.cz (J.V.); ceccio@ujf.cas.cz (G.C.); lavrentiev@ujf.cas.cz (V.L.); 5Faculty of Civil Engineering, Czech Technical University in Prague, Thakurova 7, 166 29 Prague, Czech Republic; jiri.nemecek@fsv.cvut.cz; 6Department of Physics, The University of Illinois at Chicago, Chicago, IL 60607, USA; rfklie@uic.edu

**Keywords:** Ti_2_SnC, M_2_AX, powders, thin films, STEM, nanoindentation

## Abstract

Layered ternary Ti_2_SnC carbides have attracted significant attention because of their advantage as a M2AX phase to bridge the gap between properties of metals and ceramics. In this study, Ti_2_SnC materials were synthesized by two different methods—an unconventional low-energy ion facility (LEIF) based on Ar^+^ ion beam sputtering of the Ti, Sn, and C targets and sintering of a compressed mixture consisting of Ti, Sn, and C elemental powders up to 1250 °C. The Ti_2_SnC nanocrystalline thin films obtained by LEIF were irradiated by Ar^+^ ions with an energy of 30 keV to the fluence of 1.10^15^ cm^−2^ in order to examine their irradiation-induced resistivity. Quantitative structural analysis obtained by Cs-corrected high-angle annular dark-field scanning transmission electron microscopy (HAADF-STEM) confirmed transition from ternary Ti_2_SnC to binary Ti_0.98_C carbide due to irradiation-induced β-Sn surface segregation. The nanoindentation of Ti_2_SnC thin nanocrystalline films and Ti_2_SnC polycrystalline powders shows that irradiation did not affect significantly their mechanical properties when concerning their hardness (H) and Young’s modulus (E). We highlighted the importance of the HAADF-STEM techniques to track atomic pathways clarifying the behavior of Sn atoms at the proximity of irradiation-induced nanoscale defects in Ti_2_SnC thin films.

## 1. Introduction

The MAX phases are a family of about 90+ carbides (or nitrides) synthesized up to now with a basic stoichiometry nomenclature M_n+1_AX_n_, where M is an early transition d metal (i.e., Sc, Ti, V, Cr, Zr, Nb, Mo, Hf, and Ta), A represents an element (mainly) from the IIIA or IVA groups of the periodic table (i.e., Al, Si, P, S, Ga, Ge, As, Cd, In, Sn, Tl, and Pb), and X is carbon or nitrogen. The index n can be 1, 2, or 3, so as the stoichiometry can vary, phases AMXMA (211), AMXMXMA (312), or AMXMXMXMA (413) can be formed [1]. The MAX phases have a hexagonal crystal structure (a space group D^4^_6h_, P63/mmc) with two units per cell. The cell consists of the M_6_X octahedra that alternate with a single layer of the A elements [2].

The idea of the ternary transition metal carbides referred to as the H phases, where H denotes hexagonal close-packed structures, was developed by Nowotny and Jeitschko almost 60 years ago [3]. Later, the ternary transition metal carbides were classified as the M_n+1_ AX_n_ phases [1,2,3,4] due to their unique hybrid structure with mixed covalent/ionic/metallic properties. This unusual conjunction influences their thermodynamic stability and mechanical properties and predicts that MAX phases could be highly regarded candidates for applications in extreme conditions. Some of the MAX compounds have already driven interest in nuclear engineering as materials with high potential for future fission and fusion reactors [5]. For instance, high radiation resistance was shown for the first time on titanium aluminum and titanium silicon carbides irradiated with high fluence heavy ions (Xe^+^, 6.25 × 10^15^ ions cm^−2^) [6]. The received experimental data suggested that after irradiation, the structure of the MAX compounds keeps well-ordered. This observation was acknowledged in new experiments [7] with other members of the MAX group. Titanium tin carbide (e.g., Ti_2_SnC M_2_AX), discovered already in 1963 [3], demonstrates unusual material characteristics, such as high tolerance to mechanical damage, high modulus elasticity, and good integral stability at high temperature [8,9,10,11]. Surprisingly, there is a lack of relevant data on the ion beam and/or neutron radiation tolerance of Ti_2_SnC. Perhaps this is due to the rather high cross sections for neutron-induced γ–(gamma) activation of the Sn isotopes (~0.6 b for natural Sn) with relatively long lifetimes (e.g., ~129 d for ^122^Sn + n), which means that the Ti_2_SnC may be unattractive for nuclear engineering technology. However, as a promising coating material, Ti_2_SnC may still be interesting, and it is worth studying its irradiation resistance. The Ti_2_SnC is especially synthesized using a finely dispersed powder of the Ti, Sn, and C phases, mixed in stoichiometric ratios, grounded, pressed, and sintered at high temperatures [12,13]. Other techniques, such as spark plasma sintering [14], have been invented and used for Ti_2_SnC fabrication [15]. The applied methods, however, documented that together with the Ti_2_SnC composite, some precipitates (such as TiC, Ti_6_Sn_6,_ or Sn) are also detected. It turns out that to fully transform the correct stoichiometric ratio of the Ti-Sn-C to the acceptable ternary M_2_AX phase is still a challenge.

In this research, the Ti_2_SnC thin nanocrystalline films (TNCFs) were synthesized using an unconventional low-energy ion facility (LEIF) based on ion beam sputtering combined with further a low- temperature thermal processing up to 150 °C. The Ti_2_SnC powdered polycrystalline samples (PPS) were fabricated by simplified sintering of a compressed mixture consisting of (Ti, Sn, C) elemental powders. The goal of this study is to compare the morphological and nanomechanical features of Ti_2_SnC M2AX materials prepared by using different synthetic methods. In addition, the irradiation tolerance of the Ti_2_SnC TNCFs was examined. The irradiation was carried out by a heavy Ar^+^ ion with an energy of 30 keV to the fluence of 1.10^15^ cm^−2^. We provide an experiment in the understanding of Sn atoms surface segregation and highlight the importance of aberration-corrected STEM techniques including high-angle annular dark-field detector (HAADF) to track atomic pathway clarifying the behavior of Sn atoms at the proximity of irradiation-induced nanoscale defects in Ti_2_SnC TNCFs. 

## 2. Materials and Methods

### 2.1. Synthesis of Ti_2_SnC M_2_AX PPSs

The Ti_2_SnC M_2_AX PPSs were fabricated using a simplified method of sintering raw elemental powders. The experimental setup is presented in Figure 1a.

Stoichiometric amounts of Ti (99.7%, Aldrich, powder), Sn (≥93%, Aldrich, powder), and graphite (diamond powder) as raw materials were mixed with isopropyl alcohol at a 1/0.8/0.9 molar ratio and ground in an agate mortar. After thorough grinding, 0.1 mL of a ‘pressing solution’ (ethanol solution of polyethylene glycol 400, 1% *w*/*w*) was added to the fine ground suspension and mixed thoroughly again. After evaporation of the alcohol component, the solid mixture of reagents was tightly compressed (at a specified pressure of 750 MPa) to form a pellet with a diameter of 1.3 cm. The pellet was then placed into a corundum tube of a laboratory furnace and heated at a temperature up to 1250 °C under a vacuum. The optimal annealing regime to receive the Ti_2_SnC M_2_AX PPS was determined to be following (1. step 0–1050 C–20 C/min, 2. step 1050–1150C–10C/min, 3. step 1150–1250C–5C/min, 4. step 1250C-delay 2 h). The as-obtained pellet was milled and heated again under the same regime. The optical micrographs of Ti_2_SnC PPS are presented in Figure 1a.

### 2.2. Synthesis of Ti_2_SnC TNCFs

A set of Ti_2_SnC TNCFs was prepared by ion beam sputtering (IBS), as well as controlled thermal processing. The ions were generated in the high-current ion source (duoplasmatron) placed in the LEIF (lab-made Low Energy Ion Facility assembled by NPI) of the CANAM (Center of Accelerators and Nuclear Analytical Methods) research infrastructure in the NPI Rez [16]. The LEIF facility lets us utilize different gaseous ions with energy in the range of 100 eV to 35 keV, and an ion current up to 500 μA. The beam spot size of the Gaussian-shaped ion beam was about 20 mm. In this report, the Ar^+^ ions-beam has been used (apart from singly charged Ar^+^, double-charged Ar^2+^ ions were also present, though only a fraction of a few %). The current and energy of the Ar^+^ ion have been varied to measure the optimal values for the manufacture of the titanium tin carbide. For the fabrication of the Ti_2_SnC TNCFs, highly purified targets of Ti (99.995%), Sn (99.999%), and C (99.999%), all of the MaTeck (MaTeck Material Technologie & Kristalle GmbH, Juelich, Germany) materials have been used. The specimens were placed on a metallic (Cu) stocker (a frame with an equilateral triangle shape), each on a different side of the frame, that was mounted in the sputtering chamber of the LEIF system. The schema of the LEIF sputtering is presented in Figure 1b and described in detail elsewhere [17].

The targets with a size of about 5 cm in diameter covered the dimensions of the frame, so the sputtering from the target holder itself was stopped. The holder was connected to a metallic (Cu) axis (controlled from the outside of the chamber) and cooled down forcefully (by distilled water of an external cooling system) when specimens were overheated. The holder was revolving in 3 shifts with a speed of 1 rotation per 100 s. The revolution was performed automatically using a stepper motor (Accu-Glass Products. Inc., Valencia, CA, USA) operated by a PC. In each shift of 60°, a corresponding composite (Ti, Sn, or C) was sputtered for a definite period t_phase_ to deposit a necessary amount Δ (~10^15^ cm^−2^) of the specimens’ material on the substrate. The sputtering times were defined by the deposition rates DR_phase_ of the particular composites and by the stoichiometric ratio of the Ti_2_SnC phase (2:1:1): t_Ti_ = 2Δ/DR_Ti_, t_Sn_ = Δ/DR_Sn_, t_C_ = Δ/DR_C_. An approximately 1 nm thick layer with 2Ti+lSn+lC atomic mixture was deposited during each rotation. The deposition was conducted on Si wafers or Mo TEM grids with ion energy of 25 keV and an ion current of 400 micro A. The deposition rates were held permanent and they were defined for Ti—0.85 × 10^15^ cm^−2^, Sn—9.80 × 10^15^ cm^−2^ and C—0.76 × 10^15^ cm^−2^ per min using RBS. A set of the samples was annealed at 150 °C for 24 h in a vacuum to induce interphase chemical interaction and complete formation of the stoichiometrically correct Ti_2_SnC M_2_AX phase. Samples were labeled Ti_2_SnC_AGTNCF, where AG was denoted “as-growth” thin film. The Ti_2_SnC_AGTNCF was further irradiated by 35 keV Ar^+^ ions in order to examine their irradiation-induced resistivity. Optical micrographs Ti_2_SnC TNCFs obtained by LEIF are presented in Figure 1c–e. 

### 2.3. Ion Beam Irradiation

In order to get pieces of knowledge about the radiation tolerance of the Ti_2_SnC_AGNGTFs, a heavy Ar^+^ ion with an energy of 30 keV to the fluence of 1.10^15^ cm^−2^ was applied. It is assumed that Ar^+^ ions with an energy of 30 keV generate an irradiated (damaged) area of 30 nm deep from the film surface. Using the SRIM-2013 code, the dpa value for this fluency was evaluated at 9.49 dpa (in the calculation, the density of Ti_2_SnC—6.36 g cm^−3^, displacement energy—25 keV, and ‘energy to recoil’—81 eVÅ^−1^ were considered). The Ti_2_SnC_AGTNCFs were tested by several nuclear analytical methods. The thickness was examined with a sub-nanometer precision profilometer KLA-Tencor Alpha-Step IQ Surface Profiler/, as well as by Rutherford backscattering spectrometry (RBS; lab-made, assembled in NPI). 

### 2.4. Methods of Characterization

The powder diffraction patterns of the Ti_2_SnC PPS were obtained with a PANalytical X’PertPRO MPD diffractometer (Malvern, United Kingdom) equipped with the Cu Kα tube (λ = 0.15406 nm). The diffractometer was operated at 40 kV and 30 mA with a 0.5° divergent slit coupled with a 0.1 mm receiving slit. Room temperature diffractograms were recorded in the transmission regime in the range from 5° to 85° at a 2θ step size of 0.01°. The phase composition of the measured powdered sample was calculated by the Rietveld analysis in an automatic mode of HighScore software 5.0 (Malvern, United Kingdom). 

Scanning electron microscopy (SEM) was used for the characterization and imaging of the fine surface structure of the prepared Ti_2_SnC_TNCFs and Ti_2_SnC PPS. For measurement, a JSM 6510LV system (low vacuum JEOL microscope, Jeol Ltd., Tokyo, Japan) with an acceleration voltage of 0.5–30 kV was used. For analysis, the secondary electron imaging mode (SE) was applied. 

A detailed microstructural analysis of electron diffraction, and also elemental mapping, were carried out on a high-resolution transmission electron microscope (HRTEM) JEOL JEM 3010 Jeol Ltd., Tokyo, Japan). The microscope was operated at 300 kV (using a LaB_6_ cathode; the point resolution was 1.7 Å), and it was equipped with an energy-dispersive X-ray (EDX) detector (Oxford Instruments, High Wycombe, UK) for elemental analysis, and a Gatan CCD camera (1024 × 1024 pixels) for image recording. The obtained images were analyzed using the Digital Micrograph software 3.5 package Gatan, California, USA), the EDX analysis was processed with the INCA software package (High Wycombe, UK). Electron diffraction patterns were evaluated using the ICDD PDF-2 database, Newtown Square, PA, USA [18] and ProcessDiffraction V_8.7.1. Q software 7 package (Budapest, Hungary) [19]. For the TEM analysis, a small bit of the pellet sample was crushed, dispersed in ethanol, and the obtained suspension was sonicated for 2 min. A drop of the very dilute suspension was then placed on a holey-carbon coated Cu-grid and allowed to dry by evaporation at ambient temperature.

The atomic resolution Z-contrast images of Ti_2_SnC_AGTNCFs and Ti_2_SnC_Ar^+^TNCFs were collected using the JEOL ARM200CF (Jeol Ltd., Tokyo, Japan) aberration-corrected STEM with a cold-field emission gun operated at an acceleration voltage of 80 kV. The high-angle annular dark-field (HAADF) images were acquired using an annular dark-field detector with a collection angle ranging from 90 to 175 mrad. The probe convergence semi-angle was set to 29 mrad, which yields a probe size of 1 Å at 80 kV and a probe current of 62 pA [20].

The surface topography of the Ti_2_SnC_TNCFs fabricated by LEIF was studied by atomic force microscopy (AFM) using the NTEGRA scanning probe microscope (NT-MDT Spectrum Instruments, Moscow, Russia). The AFM experiments were performed under ambient conditions using tapping mode for the acquisition of the sample surface images (AFM topography). 

The nanomechanical properties of the Ti_2_SnC_AGTNCFs, Ti_2_SnC_Ar^+^TNCFs, and Ti_2_SnC PPS were inspected by nanoindentation with a Hysitron Tribolab TI-700 Nanoindenter (Bruker Nano GmbH, Berlin, Germany) equipped with a Berkovich tip. Indentations were made at 4 distant locations (6 indentations at each), and each measurement consisted of 10 cycles with penetration depths between 20–150 nm (with a contact depth of 10–130 nm) to analyze changes induced by irradiation in the material’s properties. The Hysitron TI-700 used with Berkovich tip is capable of quantitative measurements for depths larger than 10 nm, for which the tip calibration done on fused silica standard was performed. In the calibration procedure, the tip radius is not explicitly assessed, but the contact area is evaluated before the measurements. Measurements in larger depths can be considered accurate with the accuracy of the polynomial contact area calibration function (R^2^ = 0.999).

## 3. Results and Discussion

### 3.1. Elemental Detection by Nuclear Analytical Methods

In Figure 2, both RBS and non-Rutherford spectra of Ti_2_SnC_AGTNCFs are presented together with the results obtained by simulations with SIMNRA code (performed on the Tandetron 4230 MC). It was registered that *Ti_2_SnC_ADTNCFs* were contaminated by oxygen up to a level of about 35%. However, the ratio of the Ti, Sn, and C elements keeps up the stoichiometric ratio (Ti/Sn/C~2/1/1). 

We can suggest that Ti_2_SnC_AGTNCFs were (partially) oxidized either during the deposition process (with the residual oxygen in the sputtering chamber), or through the annealing in a relatively low-level vacuum of 10^−4^ Pa [21]. Therefore, the synthetic challenge is to avoid oxidation contamination which occurs commonly when LEIF deposition is used [22].

The thickness of the Ti_2_SnC_AGTNCFs on the polished Si wafers with a size of about 1 cm^2^ was detected to be in the diapason 460–920 × 10^15^ cm^−2^. 

### 3.2. Morphology of Ti_2_SnC_AD, Ti_2_SnC_Ar^+^TNCFs, and Ti_2_SnC PPS Imaged by AFM and SEM

Figure 3 shows the AFM results obtained from analysis of the Ti_2_SnC_AGTNCFs and Ti_2_SnC_Ar^+^TNCFs taken from the sample surface area of (1 × 1) μm^2^. 

Surface profile plots were prepared along the horizontal blue lines in the AFM images (see the plots from the right panel in Figure 3a,b). It is seen that the Ti_2_SnC_AGTNCFs surface (Figure 3a) consists of the nanoparticles (NPs), which are mostly separated each from other. Different color of the NPs reflects their different height, suggesting the formation of non-uniform agglomerates during the film deposition. The latter effect results in the relatively high surface roughness, which was found to be *SR_rms_* = 3.275 nm (*SR_rms_* denotes the root mean square roughness of the surface). The details of the NPs agglomerations are seen in the magnified image of the (250 × 250) nm^2^-sized surface area (see the AFM image from the left panel), selected by the dotted-line square in the original AFM image. Analysis of the magnified image and the surface profile reveals the size distribution of the NPs, with a lateral size of 20–30 nm. Annealing at 150 °C (*T_a_* = 150 °C) and irradiation with the Ar^+^ ion beam modified the morphology of the Ti_2_SnC_Ar^+^TNCFs. Surface roughness was found to be significantly higher (*SR_rms_* = 7.057 nm) than that of Ti_2_SnC_AGTCNF. According to the magnified image (the left AFM image in Figure 3b) and the surface profile, bigger agglomerates with a lateral size of 50–90 nm appear on the Ti_2_SnC_Ar^+^TNCFs surface. The spatial density of the agglomerations is also higher, and the size distribution shifting to the larger NP size was observed. It was rather hard to obtain AFM vital data for Ti_2_SnC PPS. The polishing technique and mechanical surface treatment during sample preparation were the main obstacles for representative AFM imaging. Despite that, Appendix A shows the SEM surface morphology of the Ti_2_SnC PPS. It is found that misaligned grains and pores are formed on the Ti_2_SnC PPS surface during hydrothermal synthesis. A smooth surface and fully dense microstructure without defects can be seen in SEM micrographs of Ti_2_SnC_AGTNCFs (Appendix A). When irradiation with an Ar^+^ ion beam was applied, the small round-shaped features on the surface of the Ti_2_SnC_Ar^+^TNCFs appeared (Appendix A), suggesting that Ar^+^ ion-beam irradiation-induced blistering and/or surface defects. In order for the structure of Ti_2_SnC_AGTNCFs and Ti_2_SnC Ar^+^TNCFs to be solved, atomic-resolution bright-filed (BF) and HAADF imaging in C*s*-corrected STEM was further performed and models for atomic ordering of Ti_2_SnC_ AGTNCFs and Ti_2_SnC Ar^+^TNCFs structures were proposed.

### 3.3. Structural Analysis of the Ti_2_SnC_AGTNCF with Atomic-Resolution STEM

Figure 4 shows the aberration-corrected BF and HAADF STEM micrographs of Ti_2_SnC_AGTNCF acquired from different magnification. Two regions can be distinguished in the HAADF-STEM images in Figure 4a,b; matrix with lower intensity (labeled 1) and well-crystallized spherical NPs (labeled 2), which present higher intensity (see a yellow boxed region in Figure 4b). This contrast can be associated with the atomic weight dependence of constituted elements. It should be noted that the contrast of Sn appears brighter than the Ti since the atomic number of Sn (*Z* = 50) is larger than that of Ti (Z = 22) in the HAADF electron scattering regime. The corresponding SAED (Figure 4c) confirms a mixture of two phases: tetragonal SnO (JCPDS 06–0395 space group P4/nmm) and hexagonal Ti_2_InC (JCPDS PDF 01-089-5590, space group P6_3_/mmc). A zoomed STEM image of the matrix is displayed in Figure 4f with an incident electron beam along the [001] direction. The measured lattice fringe spacings d_(100)_ = 0.27 nm correspond to hexagonal Ti_2_SnC. HAADF STEM analysis (Figure 4g–i) further provides additional evidence of the formation of SnO (SnO-Sn^2+^). In Figure 4i, (101) atomic planes with interplanar spacing d = 0.29 nm for tetragonal SnO with growth in the *c* parameter direction on the surface of Ti_2_SnC can be seen. Therefore, a protective Ar_2_ atmosphere could avoid the oxidation of Sn to the thermodynamically more stable SnO_2_ (SnO_2_-Sn^+4^) [23,24]. A similar mechanism of SnO formation/SnO_2_ avoiding was reported upon using a protective N_2_ atmosphere for the synthesis of uniform nanocrystalline SnO layers [25,26]. 

Simultaneously performed EDS-STEM quantitative analysis (Appendix A) confirmed the same elements (Ti, Sn, C, O) as detected by RBS and non-Rutherford scattering (see Figure 2). It can be seen from the elemental distribution maps (Appendix A) that the mapping image of Sn is interconnected with the O mapping image. Such a correspondence reveals that Sn appears in oxygen-enriched regions, for example, spherical shaped NPs on the surface, and indicates that Sn can exist as an Sn-oxide rather than metal Sn [27].

### 3.4. Structural Analysis of the Ti_2_SnC_Ar^+^TNCF with Atomic-Resolution STEM

A structural anomaly in Ti_2_SnC_Ar^+^NCTF was found to be driven during Ar^+^ ion beam irradiation. Low magnification BF (Figure 5a) and HAADF-STEM (Figure 5d) micrographs revealed well-crystallized nanograins. Figure 5b,e report magnified BF and HAADF STEM images, where interconnected disc-like NPs (labeled 1) and spherical NPs (labeled 2) were observed (Figure 4b,e). 

Besides the distinct morphology, the SAED pattern (Figure 5b) taken from the yellow boxed area in Figure 5a did not match those of Ti_2_SnC. It was found that the lattice spacing and the angle of hexagonal Ti_2_SnC lattice structure undergo a fundamental transformation. For identified [001] zone axis, the main (111), (220), and (200) lattice planes for cubic Ti_0.98_C phase (i.e., space group *Fm-3m*, JCPDS PDF No. 04-004-2862) were detected. Furthermore, brighter reflections on the ED patterns were found to be extremely close to the (101) and (110) planes, matching well with an β-Sn with tetragonal symmetry (i.e., space group *I4/mmm*, JCPDS PDF No. 00-018-1380). We noticed that some diffraction spots (white boxed in SAED pattern) do become dimmers. High contrast variation in the HAADF STEM image (Figure 5e) suggests that the surface of disc-like NPs (labeled 1) is covered with smaller spherical NPs with bright contrast (labeled 2). Concerning the difference in atomic numbers of constituent elements and approached distinct Z-contrast, we could infer that Ar^+^ ion beam irradiation promoted the growth of fine spherical β-Sn NPs running along disc-like Ti_0.98_C NPs [28,29,30]. The proposed dual heterostructure that evolved under the Ar^+^ ion-beam irradiation process was further confirmed by aberration-corrected HAADF-STEM imaging and simulation of experimental SAED patterns. We discuss first the structure of disc-like NPs in Figure 6a. The FFT in the [110] orientation (inset in Figure 6a) reports the major lattice planes (111), (220), and (200) for the cubic Ti_0.98_C phase. Figure 6b shows an aberration-corrected HAADF-STEM image viewed along with the [00$\bar{1]}$ zone axis. The structure is maintained to the surface on the (001) planes as obtained from the yellow marked area in Figure 6a. At this surface, there are only bright atomic columns, suggesting an atomic arrangement expected for the lattice with the space group Fm-3m, where the Ti^2+^ ions occupy the tetrahedral sites. The structure model of Ti_0.98_C is superimposed on the HAADF- STEM image. For irradiated Ti_2_SnC_Ar^+^TNCF the atomic stacking transforms from ABABA/TiCSnCTi to ABBA//TiCCTi (Figure 6c) [21]. We could expect that the atomic arrangements change to ABBA because all Sn atoms are segregated from the Ti_2_SnC lattice and transition from M_2_AX to M_0.98_X structure has occurred. The simulated ED pattern of cubic Ti_0.98_C agrees with the experimental FFT. Simulated [001] HRTEM image at a focus value f = −440 Å and a thickness t = 19 Å in Figure 6d is in line with the experimental contrast. The Ti columns can be seen in the simulated [110] HRTEM image that the Ti position appears as bright dots. The interlayer spacing of 0.24 nm is between the (111) cubic planes in the Ti_0.98_C structure. The unit cell for the cubic Ti_0.98_C arrangement is overlaid on the simulated STEM image.

The white contrast in the HAADF-STEM images in Figure 6e suggested that spherical NP with corresponding FFT (inset in Figure 6e) resemble tetragonal β-Sn. We can index the set of lattice planes (001), (101), and (200) and identify the [001] axis (the *c* axis) as the orientation of segregated Sn. The original STEM image was filtered by applying a Fourier mask to remove the noise and obtain clearer lattice periodicity (Figure 6f). As a result, lattice fringes with a d_(110)_ spacing of 0.26 nm, consistent with the tetragonal crystalline structure of Sn (JCPDS PDF No. 00-018-1380) was obtained. The weaker lattice fringes in the background appeared due to the adjacent cubic structure of the bottom disk-like Ti_0.98_C. Here, we were able to identify the cubic Ti_0.98_C (selected in yellow regions). Alongside the ordered structure, certain line defects–dislocation lines (marked with magenta-colored arrows) and collision cascades (selected in red regions) into the Ti_0.98_C lattice were observed. These data appear to suggest that Sn segregation into small atomic clusters prefers to grow near the dislocation lines. It could be proposed that Sn clusters and irradiation-induced defect cores (dislocations and cascades) nucleated and grew together [31,32]. Simulated ED patterns of Sn along with the [110] zone axis (Figure 6g) show that d_(110)_ = 0.26 nm crystal plane in tetragonal Sn is present. The atomic structure model of Sn along (010) in Ti_2_SnC_Ar^+^NCTF is overlaid with a simulated ED pattern. 

Appendix A shows the EDS-STEM spectrum, atomic % of elements (Table inset in Appendix A), and EDS elemental mapping (Appendix A) of Ti_2_SnC_Ar^+^TNCF. We found the presence of Ti, Sn, and C elements. The Ti/Sn atomic ratio in Ti_2_SnC_Ar^+^NCTF decreases from the initial 1.17 in Ti_2_SnC_AGNCTF to the final 0.85 in Ti_2_SnC_Ar^+^TNCF. It can be seen from the mapping images of Ti and C (Appendix A) that both elements become woven together and distributed all over the film, whereas the Sn mapping image suggests that Sn is not interconnected with Ti. Such an observation can indicate that Sn separated rather than being interwoven with Ti_2_SnC structures, which was proved true from the HAADF-STEM and SAED observation. It is worthy to mention that the EDS spectra and the quantitative EDS mapping did not indicate the presence of oxygen. The absence of oxygen in Ti_2_SnC_Ar^+^NCTF may be explained by the irradiation-induced ionization effect, which is reported to be pronounced not only in weakly bound Van der Waals elements but could be also occurred in systems with stronger bonds [33,34,35]. This ionization process can change both the equilibrium state and geometry of the overall system, including the Ti_2_SnC matrix and surface SnO “etched out” the Ti_2_SnC. Subsequently, the charge distribution in SnO, as well as its binding energy, may be altered. Therefore, we can assume that the 30 keV Ar^+^ with a fluence of 10^15^ cm^−2^ could provide enough energy to overcome the binding energy of 485.57 eV of the Sn 3d_5/2_ core level of SnO, and further contribute to the fragmentation of the SnO [36]. These structural fragments could be Sn atoms and oxygen [37]. In the presence of Ar_2_, chemical rearrangement of elements into Ar^+^-O_2_ mixture and argon-oxygen ions as Ar(O_2_)$\begin{array}{c} +\\ n \end{array}$, Ar_2_(O_2_)$\begin{array}{c} +\\ n \end{array}$ maybe generated [38]. Moreover, the presence of Ar peak in the EDS spectrum is very hard to evidence (Appendix A) since its only characteristic diffraction peak at 3.0 keV [39] is overlapped with Sn shoulder or probably because the Ar peak intensity is close to the noise level of the EDS spectrum. Additionally, the EDS technique could not be applicable for the detection of Ar_n_(O_2_)$\begin{array}{c} +\\ m \end{array}$ traces. 

### 3.5. Mechanism of Irradiation-Induced Structural Transformation in Ti_2_SnC TNCFs

Although all of Sn containing ternary Ti_2_SnC M_2_AX phase is very studied, the role of Sn element is still under debate. It is well known that Sn belongs to the Carbon family, group 14 (IVA) of the periodic table. Unlike other elements in the group, the Sn exists in two different allotropes, metallic β-Sn (malleable) and nonmetallic α-Sn (brittle). Despite that β-Sn is the more common stable form, back transition process, from α-Sn to β-Sn at low temperatures of −50 °C is also well documented. This transition is called tin pest and hints at the different properties of Sn-based compounds. Apart from the recently reported ability of Sn to segregate in the early stages of crystallization and to act as heterogeneous nucleation sites for the secondary precipitated phase [40], other competing mechanisms claimed out that the Sn atoms could be activated to excessive secondary segregation into facets along with the site of the yet crystallized matrix [41]. 

The segregation of β-Sn and formation of dual β-Sn/Ti_0.98_C heterogeneous structure in irradiated Ti_2_SnC_Ar^+^TNCF was unraveled using atomic level direct experimental STEM observation (see Section 3.4). Close inspection of structures is present in the aberration-corrected HAADF images in Figure 7. The original ADF images are filtered using the annular mask tool in Digital Micrograph to remove high-frequency noise, and presented at the same magnification. Figure 7a reveals that Ti_2_SnC_AGTNCF was dislocation-free, but lattice parameters obtained by Single Crystal software (Oxford, England) based on SEAD patterns were estimated to be larger than proposed for hexagonal Ti_2_SnC (JCPDS PDF 01-089-5590, space group P6_3_/mmc). This observation suggests that residual strain in Ti_2_SnC_AGTNCF has remained during the preparation. 

The Ti_2_SnC_Ar^+^TNCF (Figure 7b) was found to have a highly distorted structure. The regions of distorted structure are indicated with yellow, blue, and red arrows. Suffering from Ar^+^ ion beam irradiation, the Sn atomic displacement phenomenon, as the first consequence of irradiation, can result in the formation of point defects in Ti_2_SnC lattice [42]. Additionally, irradiation-induced dislocations can provide channels for very fast Sn mass transport. Coming back to the thickness of the Ti_2_SnC_AGTNCF estimated to be in the diapason 460–920 × 10^15^ cm^−2^ (Section 3.1), an onset of the metamorphic layer due to melting of Sn (231.9 °C) could take place during the Ar^+^ ion-beam irradiation. For Ti_2_SnC, the migration energy barrier of Sn determined by ab initio calculation was found to be low enough (0.66 eV) to allow the self-diffusion of Sn atoms. These irradiation-induced defects can generate atomic transport in Ti_2_SnC lattice and as a sequence, an extreme case, when all Sn atoms are extracted from Ti_2_SnC could have occurred [43,44]. The STEM analysis reveals that in the region of the distorted structure indicated with a blue arrow, an isolated particle having a spacing of 0.26 nm that could be attributed to the β-Sn, is in contact with the particle, having space of 0.24 nm, that corresponded to Ti_0.98_C. Even though staring Ti_2_SnC_AGTNCF has M_2_AX stochiometric, Ar^+^ ion-beam irradiation lowered stability of the Ti_2_SnC and promoted its decomposition into Sn and Ti_0.98_C. Our observations are in line with the attempts to correlate lattice parameters’ c/a ratio with the stability of the 211 MAX phases as a function of Sn concentration. In addition, Ti_2_SnC combines a small M-atom with a large A-atom and the distortions due to the steric effect in both building blocks of MAX, octahedral and trigonal prisms, should be considered [45]. Therefore, radiation-induced dislocations and point defects in Ti_2_SnC can trigger diffusion and segregation of Sn atoms in irradiation-induced metastable Ti_2_SnC structure [46]. The rate at which the Sn concentration increases depends probably on the Ti_2_SnC NCs orientation, i.e., the precipitation of β-Sn has a specific crystallographic orientation relationship with the Ti_2_SnC matrix. In our case, β-Sn nucleated on Ti_0.98_C along the (110) planes as a result of the lattice stress, which induced a ⟨110⟩-oriented β-Sn pattern on the Ti_0.98_C surface (see Figure 7b) [47,48,49]. Concerning the above-discussed results, a model for describing the irradiation-induced behavior of Ti_2_SnC could be proposed to follow the steps: introducing of metastable Ti_2_SnC phase → spontaneously growth of Sn core at the initial stage of irradiation → interaction between Sn core and irradiation-induced defects → remove of metal β-Sn and restoring of Ti_2_SnC to equilibrium Ti_0.98_C concentration [21,50,51,52].

### 3.6. Structural Analysis of the Ti_2_SnC_PPS with HRTEM/SAED

Microstructure and phase composition of Ti_2_SnC PPS were investigated by HRTEM/SAED and XRD. HRTEM micrograph in Figure 8a demonstrates another method of preparation graded material with various particle shapes. High magnification from the red boxed region in Figure 8a confirmed plates with an average size of 200 nm each, decorated with small nanograins located on the top edge of each plate (see Figure 8b). The corresponding SAED pattern (inset of Figure 8b) depicts the well-crystallized hexagonal Ti_2_SnC with resolved (100) and (101) lattice plane (JCPDS PDF 01-089-5590, space group P6_3_/mmc) along the [110] zone axis. An amorphous layer with a thickness of 10 nm (marked with red arrows in Figure 8b) was also well recognized. As expected, the HRTEM image (Figure 8b_1_) of single nanograin confirmed lattice fringes with spacing 0.23 nm observed for Ti_2_SnC. An HRTEM micrograph in Figure 8c corresponding to the blue framed area in Figure 8a revealed particles with different morphology as compared to the plate-shaped particles. Single NC with a length of about 50 nm and anisotropic 1D growth achieved through the preferred [111] orientation can be observed. The corresponding SAED pattern (inset in Figure 8c) confirmed the well-crystallized cubic TiC_0.55_ with resolved (111) and (200) lattice plane and highly ordered lattice fringes with d_(111)_ spacing of 0.24 nm (Figure 8c_1_) observed for TiC_0.55_ with JCPDS PDF No. 04-018-5143.

Our HRTEM/SAED observations are in line with the XRD analysis of the Ti_2_SnC_PPS. As one can see in Appendix A, the XRD pattern exhibits 70.4% of a single Ti_2_SnC phase following JCPDS PDF 04-005-0037. The diffraction peaks are sharp and confirmed a sample with high crystallinity. The dominance of the highest peaks, i.e., (103) for the in-plane pattern of hexagonal Ti_2_SnC phase, is well recognized. The calculated value for NPs size by Scherrer’s formula was 120 nm [53]. In addition, there is a presence of 11.2% TiC_0.55_ JCPDS PDF 04-018-5143 phase and 9% of Sn JCPDS PDF04-004-7745. Our XRD results are consistent with those of Li et al. [13] as well as with M.W. Barsoum, [54] which confirmed that the content of Ti_2_SnC increases with increasing the temperature. Therefore, when the reaction temperature increases up to 1250 °C, the Ti_2_SnC becomes the prevailing phase. Additionally, our results confirmed that when Sn presents in the composition range lower than 10%, no stable intermetallic impurities such as Ti_3_Sn, Ti_6_Sn_5_, Ti_2_Sn, and Ti_5_Sn_3_ will be formed, and Ti_2_SnC can retain a single phase up to 70% homogeneity [55,56].

### 3.7. Nanomechanical Properties Ti_2_SnC_AG, Ti_2_SnC_Ar^+^TNCFs and Ti_2_SnC_PPS

In this section, we consider the mechanical properties of Ti_2_SnC_AGTNCF, Ti_2_SnC_Ar^+^TNCF, and Ti_2_SnC_PPS. Table 1 shows values of Young’s modulus (E) and hardness (H) calculated from contact depths of 10–40 nm using linear extrapolation to zero depth (from 10–80 nm for Ti_2_SnC_PPS samples, respectively). The reduced modulus (Er) and hardness (H) were evaluated for each loading step by the Oliver and Pharr method [57]. Young’s modulus (E) was calculated from the reduced modulus based on the assumption of the sample Poisson’s ratio of 0.24 [58]. Figure 9 and Figure 10 show results of elastic moduli and hardness in individual points represented by dots in the figures while red lines in Figure 9 and Figure 10 represent a linear fit from the 10–40 nm or 10–80 nm region on respective samples. The non-constant trend indicates an influence of the harder substrate for larger penetration depths meaning the true surface properties in the sub-10 nm region can be even lower. Extrapolated values to zero depth are theoretical surface characteristics not influenced by the substrate effects on Ti_2_SnC_AG/Ar^+^TNCF samples or structural effects in the case of the bulk samples Ti_2_SnC_PPS pristine (non-irradiated) (Figure 11) and Ti_2_SnC_PPS irradiated (Figure 12). Nevertheless, any comparison of pristine and irradiated samples made from the results holds. The values of the substrate Si/(001) wafer as reference material are also included [17]. 

The data for the irradiated Ti_2_SnC_Ar^+^TNCF shows commensurable hardness with Ti_2_SnC_AGTNCF, which means that even irradiated with an Ar^+^ ion beam, this material may have a high resistance to plastic straining. It seems logical to predict the role of Sn in affecting the mechanical properties of the irradiated film. It was recently reported [58] that a negligible amount of 0.1 at.% of Sn in binary Al/Cu alloys could enhance the hardening of the resultant Al/Cu/Sn material even in low temperatures (100–200 °C). Our EDS analysis finds out ~20 at.% β-Sn in Ti_2_SnC_Ar^+^TNCF. Therefore, our results are in line with the statement that the concentration of Sn is the most liable concerning the hardness properties of the materials. On the other hand, more significant elastic modulus increases than those for the Ti_2_SnC_AGTNCF were observed. Based on the HAADF- STEM results, we could assume that the microsegregation of Sn atoms on the Ti_0.98_C surface could involve local shear strain [59,60]. Local modulation of the Ti_2_SnC_Ar^+^TNCF structure with point defects/voids (marked with red arrows in Appendix A–d) is well established [61]. Probably due to a low degree of surface defects and low concentration of microsegregation conducted by Sn, the elastic modulus of the Ti_2_SnC_Ar^+^TNCF is not seriously affected. The nanoindentation results for Ti_2_SnC_AGTNCF and Ti_2_SnC_Ar^+^TNCFs are graphically presented in Figure 9 and Figure 10. The hardness value and Young’s modulus for Ti_2_SnC_PPB are higher when compared to those obtained forTi_2_SnC_AGTNCF and Ti_2_SnC_Ar^+^TNCFs (Table 1). Additionally, Young’s modulus for Ti_2_SnC_PPB irradiated is higher than those of Ti_2_SnC_AGTNCF and Ti_2_SnC_Ar^+^TNCFs. The Young’s modulus and hardness for Ti_2_SnC_PPB (pristine) and Ti_2_SnC_PPB irradiated are graphically presented in Figure 11 and Figure 12, respectively. Although it is hard to generalize, the reason for such a difference between mechanical properties of bulk and thin films may be a dependency on material properties. For instance: (i) the Ti_2_SnC_AGTNCF and Ti_2_SnC_Ar^+^TNCF are thin enough, which can lead to their higher defect density when correlated with bulk Ti_2_SnC PPS, (ii) preferred orientation of grains in a sputter-deposited *c*-axis oriented Ti_2_SnC films. We can assume that irradiation could maximize the coupling between grains that can reflect the degree of texture and mechanical properties as well [62,63], (iii) differences in grain size between Ti_2_SnC_AGTNCF (~13–14 nm) and Ti_2_SnC_Ar^+^TNCF (~15–16 nm), and Ti_2_SnC_PPB (grains size ≥100 nm) (see the size distribution for Ti_2_SnC_AG and Ti_2_SnC_Ar^+^TNCF in Appendix A obtained by ImageJ software (Madison, WI, USA)) [64], and Section 3.6. Surprisingly, the particle size distribution of the irradiated Ti_2_SnC_Ar^+^NCTF was found to follow the same order as before irradiation (see Appendix A), probably due to variation of the phase composition in Ti_2_SnC_Ar^+^TNCF. This result suggested that non-only thin nanocrystalline films with an average size below 20 nm could be considered as effective materials with enhanced radiation damage tolerance, but powdered polycrystalline Ti_2_SnC could be considered as a promising resistant material when irradiated with Ar^+^ ion beam, the fluence of which does not go over 10^15^ cm^−2^ [65,66].

## 4. Conclusions

In conclusion, the microstructure and mechanical properties of Ti_2_SnC TNCFs, synthesized by an unconventional low-energy ion facility (LEIF) based on Ar^+^ ion beam sputtering of the Ti, Sn, and C targets have been investigated. Combining high-resolution HAADF-STEM analysis with simulations of SAED patterns, observed that Ti_2_SnC_AGTNCFs coexist with SnO due to oxidation of Sn during the preparation process. A significant microstructural instability was observed after irradiation of the Ti_2_SnC_AGTNCFs with Ar^+^ ion beam having an energy of 30 keV and fluence of 1.10^15^ cm^−2^. The results from simulated SEAD patterns are compatible with experimental HAADF-STEM analysis and have suggested the existence of a heterostructure composed of binary Ti_0.98_C carbide and metallic β-Sn, which could be attributed to the irradiation-induced instability of the ultrathin Ti_2_SnC film. In addition, Ar^+^ ion-beam irradiation-induced dislocation and point defects can provide channels for very fast Sn mass transport. The analysis by nanoindentation showed that the irradiated Ti_2_SnC TNCFs and irradiated Ti_2_SnC_PPS exhibited promising Young’s modulus and hardness even for the locally disordered structure in Ti_2_SnC TNCFs. This fact opens the possibility of exploiting the β-Sn/Ti_0.98_C structure as a composite where *a* harsh radiation environment could have occurred.HRTEM/SAED observations and XRD analyses of the Ti_2_SnC_PPS documented 70.4% of a single Ti_2_SnC phase. The calculated value for NPs size by Scherrer’s formula was estimated to be 120 nm. The presence of 9% of Sn avoided the formation of stable intermetallic impurities (Ti_3_Sn, Ti_6_Sn_5_, Ti_2_Sn, and Ti_5_Sn_3_) in Ti_2_SnC_PPS. The Ti_2_SnC_PPS irradiated yield the lowest hardness (H) when compared with Ti_2_SnC_PPS and unirradiated and irradiated Ti_2_SnC TNCFs. Probably, the low degree of nano crystallinity and tendency to agglomeration upon irradiation can contribute to the surface hardness of polycrystalline bulk materials.The approach here presented may be extendable to other M_2_AX nanostructured materials, and can keep attention for material science applications ranging from protective nanocoating films, ion-beam irradiation resistant parts for nuclear applications and nanoceramics, to their utilization as a precursor for MX phases.

## Figures and Tables

**Figure 1 nanomaterials-12-00307-f001:**
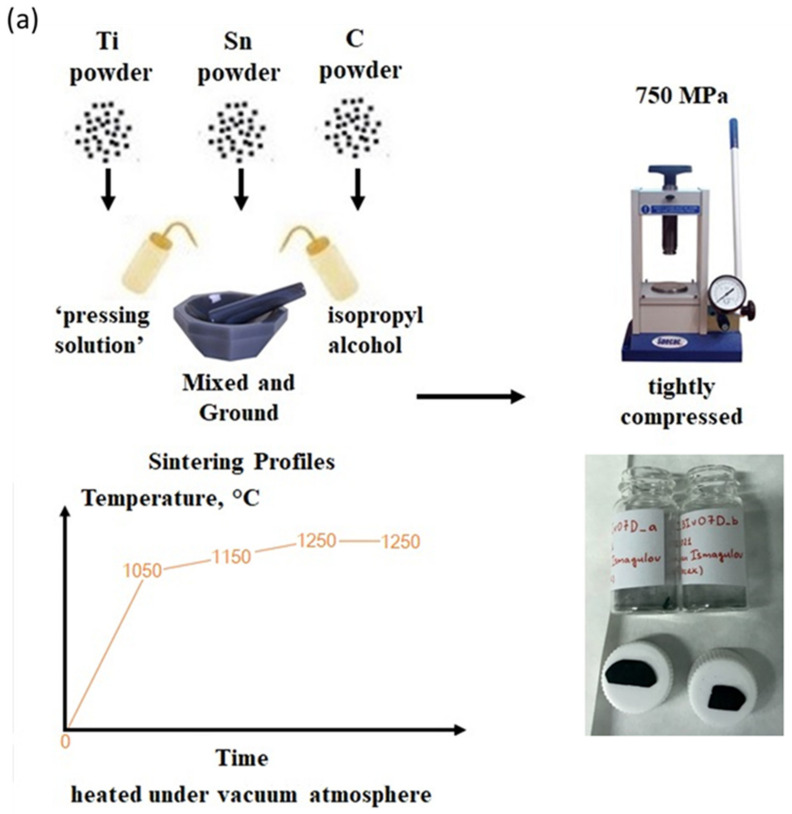
Schematic drawing of the Ti_2_SnC synthesis (**a**) Ti_2_SnC PPS by sintering of raw elemental powders with optical micrographs of Ti_2_SnC PPS at low and high magnification. (**b**) Ion beam sputter deposition setup by LEIF for the synthesis of Ti_2_SnC TNCFs. (**c**) Optical micrographs Ti_2_SnC TNCFs by LEIF. (**d**) High magnification from the blue point marks the area where an optical micrograph of a single area for unirradiated Ti_2_SnC_AGTNCFs was acquired. (**e**) High magnification from the red point mark the area where an optical micrograph of a single area for irradiated Ti_2_SnC_Ar^+^ TNCFs was acquired.

**Figure 2 nanomaterials-12-00307-f002:**
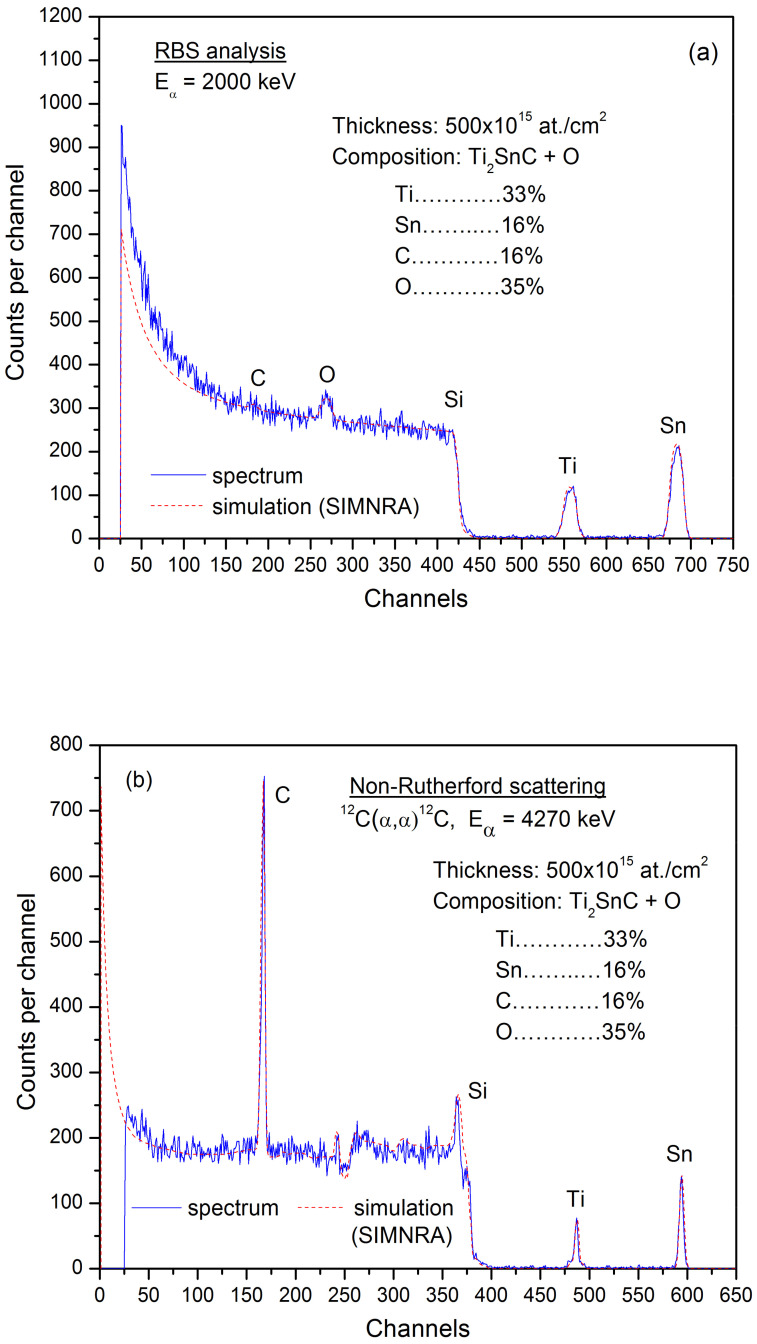
(**a**) RBS spectra of the Ti_2_SnC_AGTNCFs. (**b**) non-Rutherford scattering spectra of Ti_2_SnC_AGTNCFs by simulation of the energy spectra using the SIMNRA code.

**Figure 3 nanomaterials-12-00307-f003:**
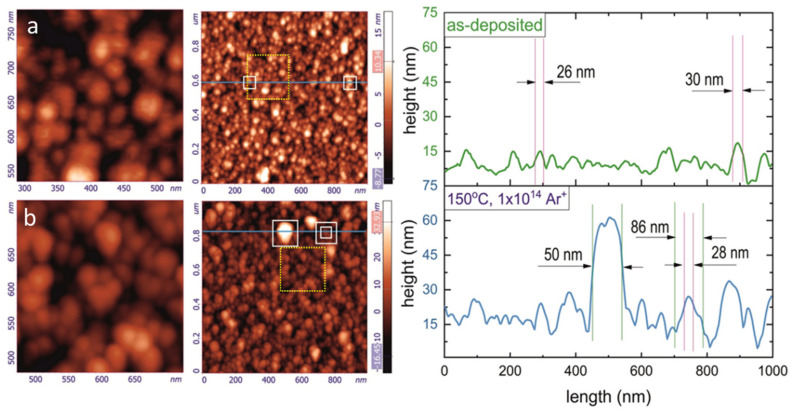
AFM characterization of the Ti_2_SnC_AGTNCF and Ti_2_SnC_Ar^+^TNCF. (**a**) AFM topography of Ti_2_SnC_AGTNCF and image magnified part from the yellow dash square (right and left images, respectively) with its corresponding surface profile. (**b**) AFM topography of the surface profile of Ti_2_SnC_Ar^+^TNCF and the image magnified part from the yellow dashed square (right and left images, respectively) with its corresponding surface profile.

**Figure 4 nanomaterials-12-00307-f004:**
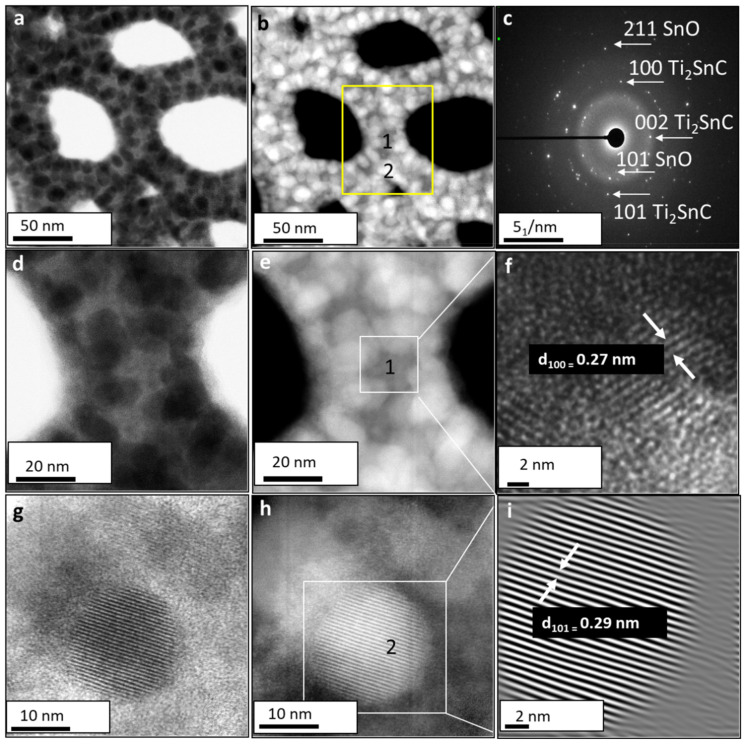
Aberration-corrected STEM images of unirradiated Ti_2_SnC_AGTNCF. (**a**) Low magnification BF-STEM image and (**b**) HAADF-STEM image where yellow boxed area outlines matrix with lower intensity (labeled 1) and brighter grains (labeled 2). (**c**) Corresponding SAED pattern showing atomic planes of SnO and Ti_2_SnC. (**d**) High magnification BF-STEM image and (**e**) HAADF_STEM image showing zoomed matrix. (**f**) High magnification from the white marked area in (**e**) where lattice fringe spacing d_(100)_ = 0.27 nm corresponds to hexagonal Ti_2_SnC was indexed. (**g**) high magnification BF-STEM image and (**h**) HAADF_STEM image showing zoomed grain with higher intensity. (**i**) High magnification from the white marked area in (**h**) where lattice fringe spacing d_(101)_ = 0.29 nm corresponds to tetragonal SnO was indexed.

**Figure 5 nanomaterials-12-00307-f005:**
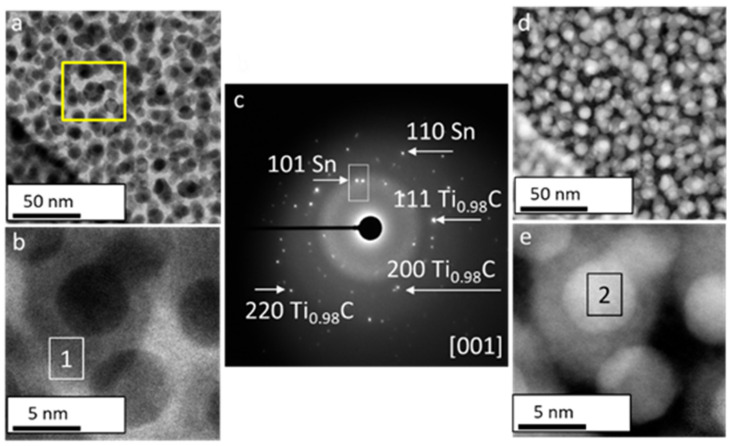
Aberration-corrected STEM images of irradiated Ti_2_SnC_Ar^+^TNCF. BF and HAADF- STEM images of Ti_2_SnC_Ar^+^TNCF (**a**) low magnification BF-STEM image and (**b**) BF-STEM high magnification of single Sn particle. (**c**) SAED from yellow boxed region in (**a**) showing atomic planes of metallic Sn and Ti_0.98_C. (**d**) HAADF-STEM image at low magnification and (**e**) HAADF-STEM image at high magnification of Sn particle appeared with bright contrast because of the contribution of scattered electrons to the Sn image.

**Figure 6 nanomaterials-12-00307-f006:**
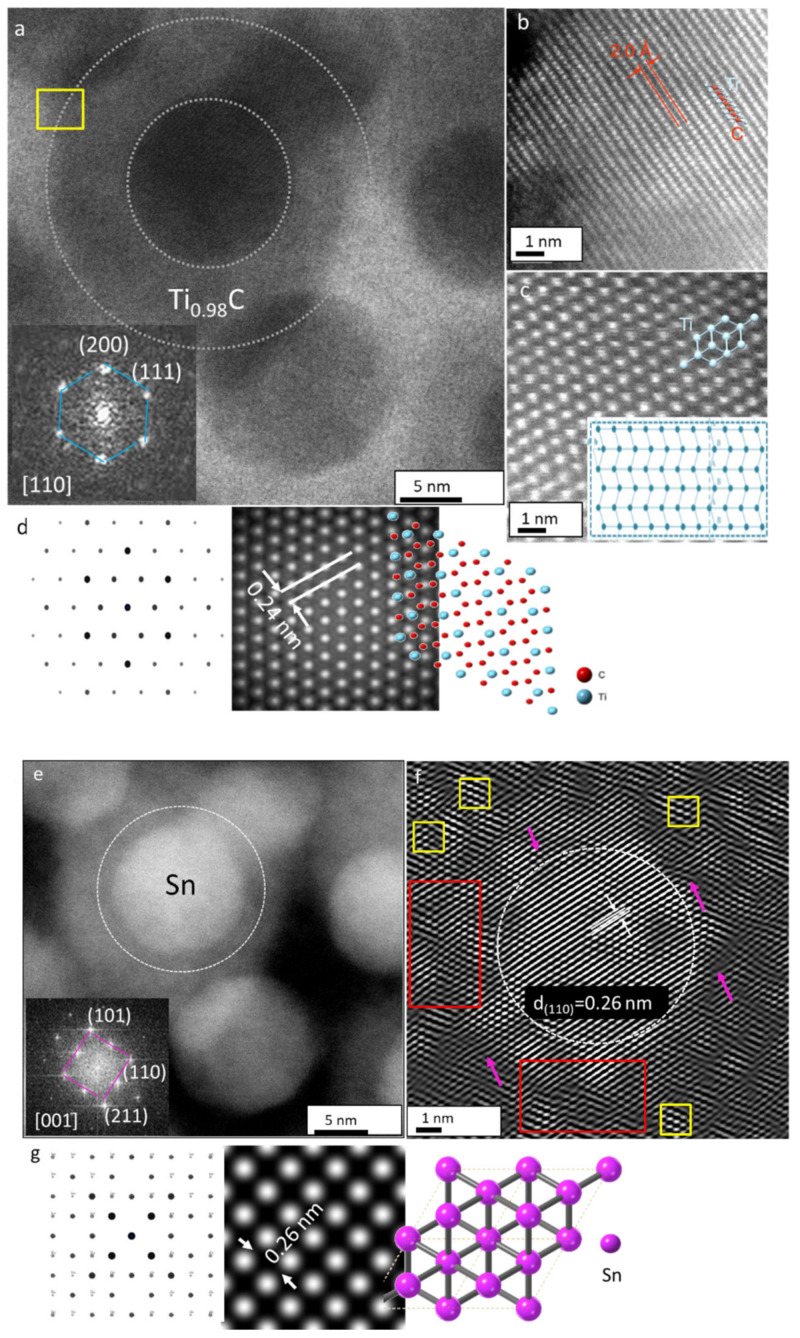
Crystalline structures of Ti_2_SnC_Ar^+^TNCF. (**a**) Aberration-corrected BF-STEM image with gray dashed lines outlines the disk-like grain as it degrades to Ti_0.98_C. Inset: FFT phase diagrams of the corresponding STEM image. (**b**) HAADF-STEM image of the yellow boxed area in disk-like grain in (**a**) with clearly resolved atomic columns. The lattice spacing is measured as 2.0 Å. The structure model of Ti_0.98_C is superimposed on the STEM image. (**c**) Atomic stacking transformation from ABABA in Ti_2_SnC to ABBA in Ti_0.98_C. (**d**) Simulated SAED patterns of Ti_2_SnC_Ar^+^TNCF structure. Transformation to the Ti_0.98_C configuration with an interlayer spacing of 0.24 nm between the (111) planes due to the escape of Sn has occurred. The unit cell for Ti_0.98_C composition with the cubic arrangement is overlaid on the simulated SAED image. Blue and red atoms refer to Ti and C, respectively. (**e**) aberration-corrected HAADF-STEM image with a white dashed line to outline the spherical NP as it segregates to β-Sn. Inset: FFT phase diagrams of the corresponding STEM image. (**f**) The corresponding HAADF-STEM image with clearly resolved atomic columns of tetragonal β-Sn. The lattice spacing is measured as 0.26 nm. (**g**) Simulated SAED patterns of the tetragonal β-Sn structure. Single Sn grain with an interlayer spacing of 0.26 nm between the (110) planes is observed. The unit cell for β-Sn composition with the tetragonal arrangement is overlaid on simulated SAED image: magenta atom refers to Sn.

**Figure 7 nanomaterials-12-00307-f007:**
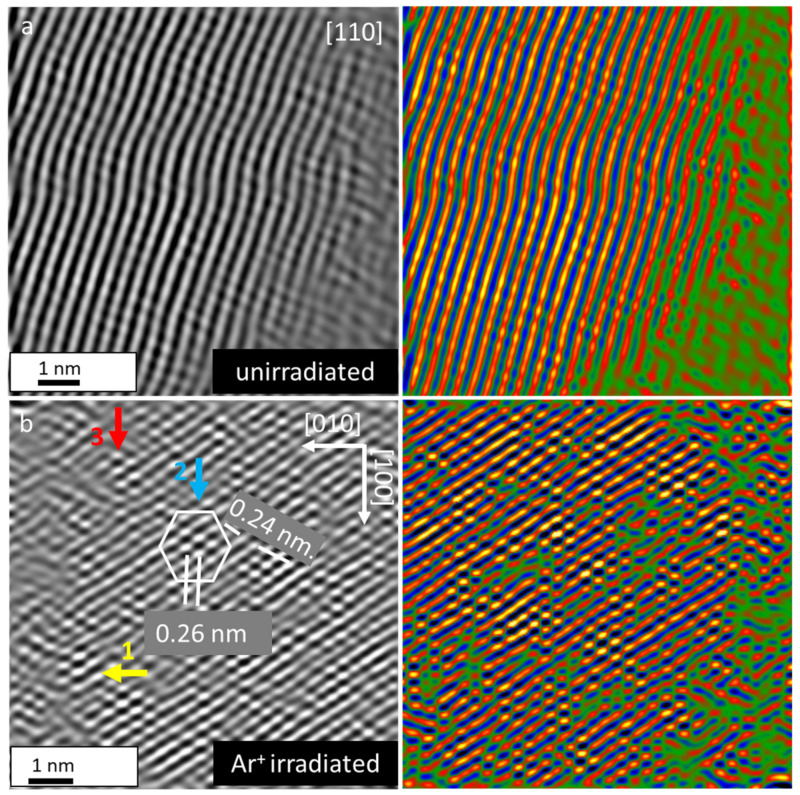
Aberration-corrected HAADF image of (**a**) Ti_2_SnC_AGTNCF along the [110] direction and (**b**) Ti_2_SnC_Ar^+^TNCF. The regions of distorted structure are indicated by yellow, blue and red arrows. The atomic arrangements of β-Sn with spacing d_(110)_ of 0.26 nm interacted with Ti_0.98_C with spacing d_(111)_ of 0.24 nm are well recognized. Corresponding filtered images to (**a**) and (**b**) by annular mask tool in Digital Micrograph are presented at the same magnification.

**Figure 8 nanomaterials-12-00307-f008:**
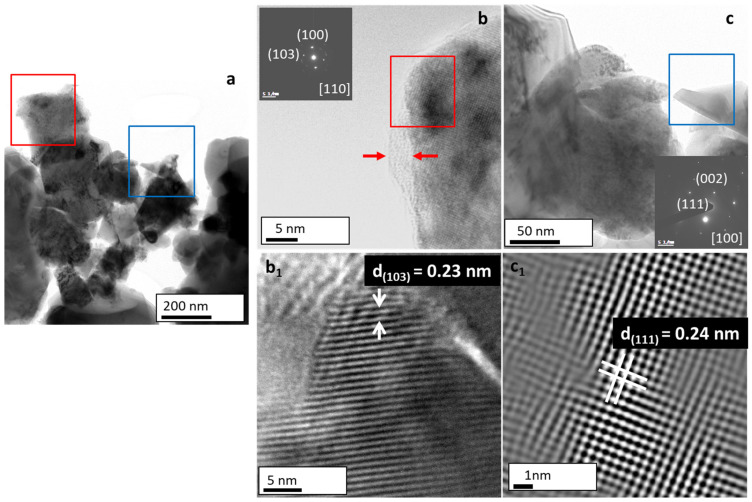
HRTEM study of Ti_2_SnC PPS. (**a**) Representative low magnification micrograph of Ti_2_SnC. (**b**) High magnification of red boxed region in (**a**) and SAED pattern with the (110) and (211) planes observed for cubic Sn. (**c**) High magnification of blue boxed region in (**a**) and SAED pattern with the (002) and (100) planes observed for hexagonal Ti_2_SnC. (**b_1_**) The magnified HRTEM image of the selected grain in red boxed area in (**b**) with lattice spacing matching these indexed in SAED pattern and (**c_1_**) high magnification image of the selected grain in blue boxed area of Ti_2_SnC in (**c**) with lattice spacing matching these indexed in SAED pattern.

**Figure 9 nanomaterials-12-00307-f009:**
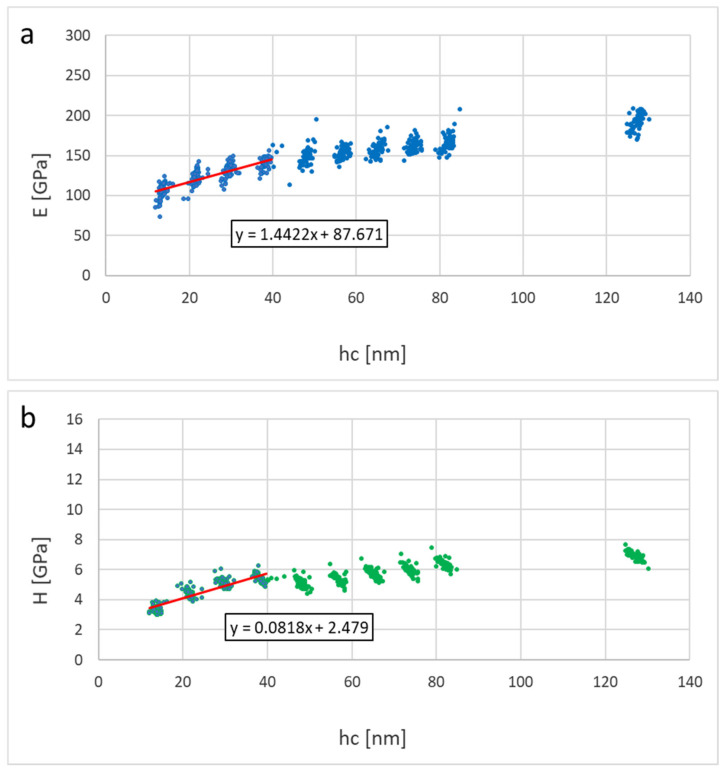
Results for Ti_2_SnC_AGTNCF (**a**) Young’s modulus vs. contact depth, (**b**) hardness vs. contact depth.

**Figure 10 nanomaterials-12-00307-f010:**
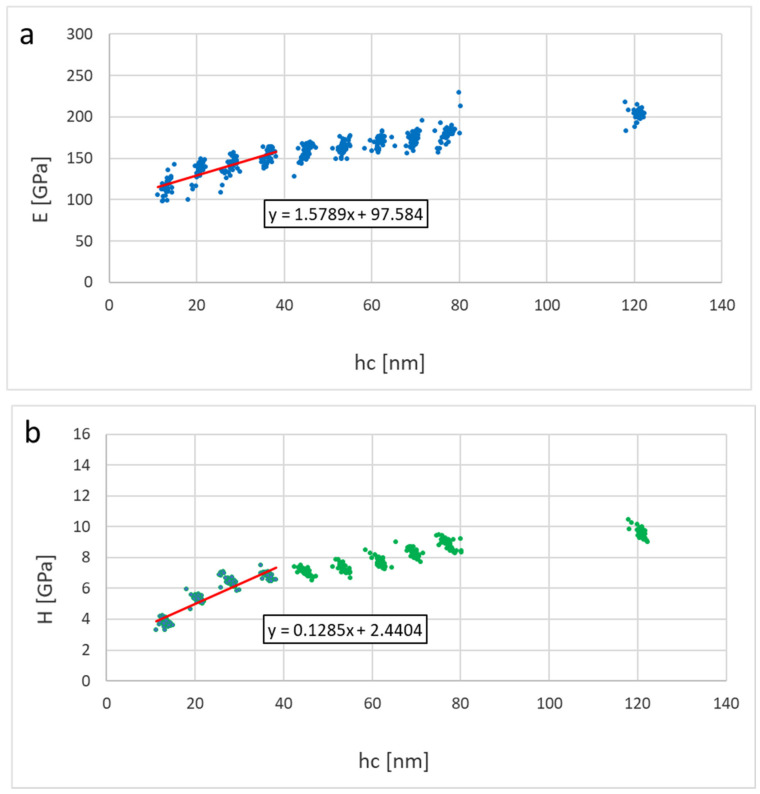
Results for Ti_2_SnC_Ar^+^TNCF (**a**) Young’s modulus vs. contact depth, (**b**) hardness vs. contact depth.

**Figure 11 nanomaterials-12-00307-f011:**
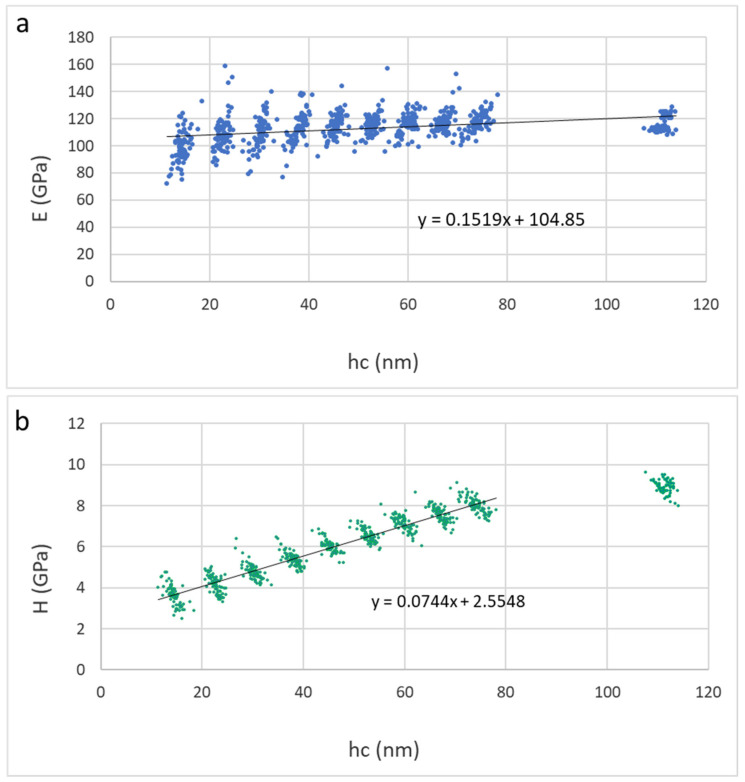
Results for Ti_2_SnC_PPS pristine (**a**) Young’s modulus vs. contact depth, (**b**) hardness vs. contact depth.

**Figure 12 nanomaterials-12-00307-f012:**
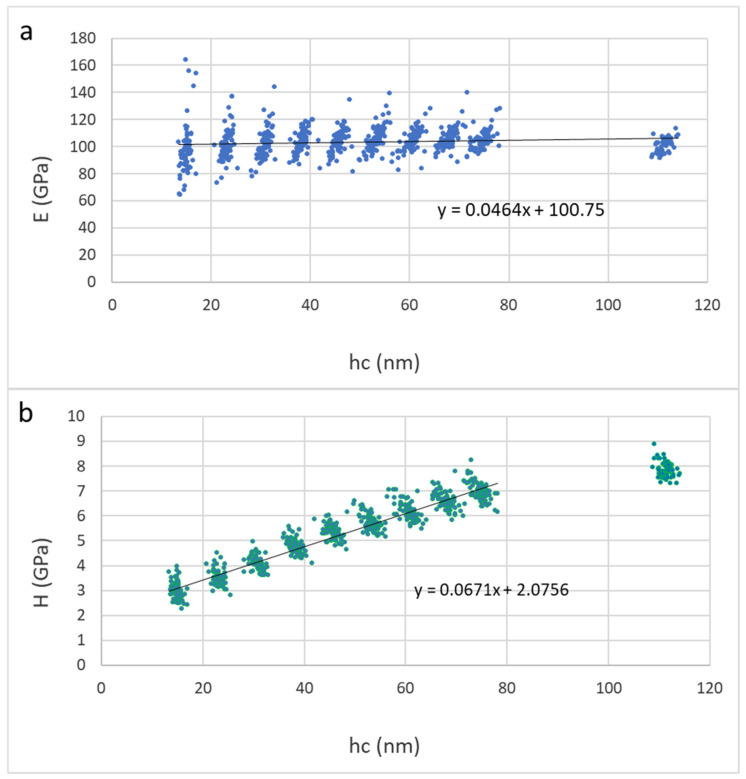
Results for Ti_2_SnC_PPS irradiated (**a**) Young’s modulus vs. contact depth, (**b**) hardness vs. contact depth.

**Table 1 nanomaterials-12-00307-t001:** Young’s modulus (E) and hardness (H) (values of linear fit at zero depth) of Ti_2_SnC_AGTNCF (film), Ti_2_SnC_Ar^+^TNCF (film), Ti_2_SnC_PPS pristine (bulk), Ti_2_SnC_PPS irradiated (bulk), and Si substrate.

Sample	E(GPa)	H(GPa)
Ti_2_SnC_AGTNCF (film)	87.7	2.48
Ti_2_SnC_Ar^+^TNCF (film)	97.6	2.44
Ti_2_SnC_PPS pristine (bulk)	104.9	2.56
Ti_2_SnC_PPS irradiated (bulk)	100.8	2.08
Si substrate, ref. [17]	166.6	15.3

## Data Availability

MDPI Research Data Policies at https://www.mdpi.com/ethics (accessed on 12 January 2022).

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
