# Peer review of "The Key Role of Tin (Sn) in Microstructure and Mechanical Properties of Ti2SnC (M2AX) Thin Nanocrystalline Films and Powdered Polycrystalline Samples"

_nanomaterials, 2022, doi:10.3390/nano12030307_

Round 1
Reviewer 1 Report
Article: “The key role of tin (Sn) in microstructure and mechanical properties of Ti2SnC (M2AX) thin nanocrystalline films and powdered polycrystalline samples”
Two synthesis methods were applied to compare the morphological and nanomechanical features of Ti2SnC M2AX materials. Production of material with enhanced properties is quite an important topic, as listed in the abstract. This work aims to study layered ternary Ti2SnC carbides that are expected to bridge the gap between properties of metals and ceramics, and as authors note, “it turns out that to fully transform the correct stoichiometric ratio of the Ti-Sn-C to the acceptable ternary M2AX phase is still a challenge”.
The study gives important information for researchers of Ti2SnC materials, and the presented approach could be extended to other M2AX nanostructured materials.
The article is recommended for publication after minor revision.
Lines 20-31, lines 79-89, etc: Different font sizes are used, which complicates reading
Line 24: The abbreviation HAADF-STEM is mentioned for the first time and should be described as follows: High-angle annular dark-field scanning transmission electron microscopy (HAADF-STEM)
Line 25-27: Probably, it would be better to paraphrase the sentence for better understanding.
Line 30: HAADF abbreviation has been already mentioned above.
Figure 1A has low resolution and needs to be refined, and maybe it would be even better to make a separate Figure from Fig.1A.
Figure 4 f, j – the length of the segment between the lines is 5 times shorter (and should be about 1nm) than the size indicated by the scale bar, which length is 5nm. Which size is wrong, the scale bar size or d-size=0.27nm?
Line 112: there is a misprint, 1. spet instead of 1. step.
Line142-144: why DRphase is written in bold?
Line 163: Please specify the model of the sub-nanometer precision profilometer.
Line 465: Probably it should be written: another method instead of “that our method”
Line 525: a misprint, Fig. S6a-d
Line 536: why it has written Fig. s6 instead of Fig. 6? the same is in line 432 Fig. s5
Figure 9a: Probably it would be better to list it as a Table, not a Figure?
Line 565: It would be beneficial to write measured hardness values (from Figure 9A) in conclusions comparing different methods.
Figure 3: C and D letters are not marked on images.
Author Response
We would like to thank the reviewers for their thoughtful comments and efforts towards improving our manuscript. In the following, we highlight the concerns of reviewers and our effort to address these concerns. We then address comments specific to each reviewer below.
REVIEWER No.1
We thank the referee for the careful and insightful review of our manuscript. We address all of the concerns of the reviewer here.
Article: “The key role of tin (Sn) in microstructure and mechanical properties of Ti2SnC (M2AX) thin nanocrystalline films and powdered polycrystalline samples”
Two synthesis methods were applied to compare the morphological and nanomechanical features of Ti2SnC M2AX materials. Production of material with enhanced properties is quite an important topic, as listed in the abstract. This work aims to study layered ternary Ti2SnC carbides that are expected to bridge the gap between properties of metals and ceramics, and as authors note, “it turns out that to fully transform the correct stoichiometric ratio of the Ti-Sn-C to the acceptable ternary M2AX phase is still a challenge”.
The study gives important information for researchers of Ti2SnC materials, and the presented approach could be extended to other M2AX nanostructured materials.
The article is recommended for publication after minor revision.
Lines 20-31, lines 79-89, etc: Different font sizes are used, which complicates reading
We checked the fond and the size in the whole manuscript. We adjusted them to be according to the Nanomaterials template.
Line 24: The abbreviation HAADF-STEM is mentioned for the first time and should be described as follows: High-angle annular dark-field scanning transmission electron microscopy (HAADF-STEM)
Thank you! We corrected the line according to the reviewers´ suggestions.
Line 25-27: Probably, it would be better to paraphrase the sentence for better understanding.
Thank you for this comment. We reconsidered and update the abstract. Please, check the text in the revised manuscript.
Line 30: HAADF abbreviation has been already mentioned above.
Thank you! The reviewer is right! We provided corrections.
Figure 1A has low resolution and needs to be refined, and maybe it would be even better to make a separate Figure from Fig.1A.
Thank you! We accepted the suggestion from the reviewer and separated Fig.1 into two different parts better described two synthetic methods. Please, check the updated Fig. 1 in the revised manuscript.
Figure 4 f, j – the length of the segment between the lines is 5 times shorter (and should be about 1nm) than the size indicated by the scale bar, which length is 5nm. Which size is wrong, the scale bar size or d-size=0.27nm?
We appreciate this remark of the reviewer. The d size is correct to be indexed as 0.27 nm. So, the scale bar in Fig.4 f-j was wrong. We corrected it properly. Check to please the updated Fig. 4.
Line 112: there is a misprint, 1. spet instead of 1. step.
We corrected the misprint.
Line142-144: why DRphase is written in bold?
We corrected this omission.
Line 163: Please specify the model of the sub-nanometer precision profilometer.
The model of the profilometer is KLA-Tencor Alpha-Step IQ Surface Profiler. We added this information in the manuscript, see section 2.3 Ion beam irradiation.
Line 465: Probably it should be written: another method instead of “that our method”
We corrected this line according to the reviewers´ suggestions.
Line 525: a misprint, Fig. S6a-d
Thank you for this remark. We changed this line properly as “red arrows in Fig. 5a-d”.
Line 536: why it has written Fig. s6 instead of Fig. 6? the same is in line 432 Fig. s5
Line 536 referred to Fig. S6 in the Suppl Info file to the manuscript, that is why is marked as Fig. S6.
The same is with Fig.S5.
Figure 9a: Probably it would be better to list it as a Table, not a Figure?
Yes, we agree with the reviewer and accepted this suggestion. We separated Fig. 9a as a Table (see Table 2 in the revised manuscript). Also, we reconsidered section 3.4. Nanomechanical properties Ti2SnC_AG, Ti2SnC_Ar+TNCFs and Ti2SnC_PPS.
Please, check the updated text, Table 1 and Figs. 9,10,11 and 12.
Table 1. Young's modulus and hardness (values of linear fit at zero depth)
|
E(GPa) |
H(GPa) |
|
|
Ti2SnC_AGTNCF (film) |
87.7 |
2.48 |
|
Ti2SnC_Ar+TNCF (film) |
97.6 |
2.44 |
|
Ti2SnC_PPS pristine (bulk) |
104.9 |
2.56 |
|
Ti2SnC_PPS irradiated (bulk) |
100.8 |
2.08 |
|
Si substrate |
166.6 |
15.3 |
Line 565: It would be beneficial to write measured hardness values (from Figure 9A) in conclusions comparing different methods.
No other methods than nanoindentation are currently available for the estimation of such thin films.
Figure 3: C and D letters are not marked on images.
We revised Fig. 3 and did corrections according to reviewers´ remarks

Reviewer 2 Report
In the present work the authors prepared Ti2SnC materials by different methods, and resulting microstructural characteristics and mechanical properties are further characterized and evaluated. The results are interesting, and would provide some insights into the fabrication of functional coating with comprehensive properties. The paper can be accepted for publication after the authors address below concerns.
- The authors tried to exam the irradiation-induced resistivity of Ti2SnC nanocrystalline thin films by LEIF method, however, only one energy and one fluence are considered, which means a case study. The authors need to comment how the irradiation parameter influence the microstructure and resistance.
- In the indentation test, the utilized penentation depth is between 20-100 nm, which is very challenging for a stable measurement. While 6 indentation tests were made for each location, the error bar should be provided. Furthermore, what’s the tip radius of utilized Berkovich tip?
- 9(b) and (c) tried to present the mechanical properties of two types of irradiated surface as well as the pristine surface. However, it is impossible to distinct the exact properties for Ti2SnC_AG and Ti2SnC_Ar+NGTF from the figure.
- The authors mentioned that “The hardness values for Ti2SnCAGTNCF and Ti2SnCAr+TNCFs are notably higher when compared to those obtained for Ti2SnC_PPB.” However, the mechanical properties of Ti2SnC_PPB value are not provided.
- Recent work (Int. J. Extrem. Manuf. 2, 012004 (2020)) on the functionalizing surface by coating should be referred.
Author Response
REVIEWER No.2
In the present work the authors prepared Ti2SnC materials by different methods, and resulting microstructural characteristics and mechanical properties are further characterized and evaluated. The results are interesting, and would provide some insights into the fabrication of functional coating with comprehensive properties. The paper can be accepted for publication after the authors address below concerns.
- The authors tried to exam the irradiation-induced resistivity of Ti2SnC nanocrystalline thin films by LEIF method, however, only one energy and one fluence are considered, which means a case study. The authors need to comment how the irradiation parameter influence the microstructure and resistance.
In the current work, the radiation tolerance of the prepared Ti2SnC thin film was tested only for Ar ions with an energy of 30 keV and a fluence of 10e15 cm-2. Similar studies were performed by the group also on another thin layer of the MAX phase, where instead of Sn element was In (1,2). And it turned out, and even this initial experiment with Ti2SnC confirmed it, that the fluence around 10e15 cm-2 is for the heavy keV ions a level where it sets a gradual change in structural parameters and related material properties (due to the role of the A element, as also shown by microscopic and nanoindentation analyzes in this study).
In principle, it can be stated that Ti2SnC thin films are radiation-resistant upto the fluence of 10e15 cm-2. At lower fluences, there are almost no significant changes in the microstructure, and no significant changes were also observed in nanoindentation (neither in the Ti2InC MAX phase, which was studied more intensely). So, due to the complexity of the study, it was agreed that in the first experiment we would focus on that critical level of 10e15 cm-2 only. From the experiments performed with Ti2InC with Ar and other ions also with higher energies and a larger range of fluences, we know that both types of MAX phases prepared in the form of thin films behave similarly to ion irradiation. Above the critical fluence of 10e15 cm-2, there was observed in the case of Ti2InC dramatic manifestations, where the element A almost completely sublimated and the MAX phase collapsed. Thus, both these composites are therefore interesting - up to the level of 10e15 cm-2 they can be considered as excellent materials for harsh environments with high radiation, but above this critical level they are completely unsuitable.
1) S. Bakardjieva, G. Ceccio, J. Vacik, L. Calcagno, A. Cannavò, P. Horak, V. Lavrentiev, J. Nemecek, A. Michalcova, R. Klie, Surface morphology and mechanical properties changes induced in Ti3InC2 (M3AX2) thin nanocrystalline films by irradiation of 100 keV Ne+ ions, Surface and Coatings Technology 426 (2021) 127775 (13 pages).
- Bakardjieva, P. Horak, J. Vacik, A. Cannavò, V. Lavrentiev, A. Torrrisi, A. Michalcova, R. Klie, X. Rui, L. Calcagno, Jiri Nemecek and G. Ceccio, Effect of Ar+ irradiation of Ti3InC2 at different ion beam fluences, Surface and Coatings Technology 26 (2020) art. 125834.
- In the indentation test, the utilized penetration depth is between 20-100 nm, which is very challenging for a stable measurement. While 6 indentation tests were made for each location, the error bar should be provided. Furthermore, what’s the tip radius of utilized Berkovich tip?
The used Hysitron TI-700 with Berkovich tip is capable of quantitative measurements for depths larger than 10 nm for which the tip calibration done on fused silica standard was performed. In the calibration procedure, the tip radius is not explicitly assessed but the contact area is evaluated prior to the measurements. Measurements in larger depths can be considered accurate with the accuracy of the polynomial contact area calibration function (R2=0.999).
- 9(b) and (c) tried to present the mechanical properties of two types of irradiated surface as well as the pristine surface. However, it is impossible to distinct the exact properties for Ti2SnC_AG and Ti2SnC_Ar+NGTF from the figure.
We thank you the reviewer for this remark. We revised all figures in section 3.4 Nanomechanical properties Ti2SnC_AG, Ti2SnC_Ar+TNCFs and Ti2SnC_PPS and upload new figures (Figures 9, 10, 11 and 12) with better resolution.
The authors mentioned that “The hardness values for Ti2SnCAGTNCF and Ti2SnCAr+TNCFs are notably higher when compared to those obtained for Ti2SnC_PPB.” However, the mechanical properties of Ti2SnC_PPB value are not provided.
We thank you the reviewer for this remark. In the revised manuscript we provide values for the mechanical properties of Ti2SnC_PPB material. The text in the manuscript was changed properly.
Please, check table 2 and new figures in section 3.4 Nanomechanical properties Ti2SnC_AG, Ti2SnC_Ar+TNCFs and Ti2SnC_PPS
Recent work (Int. J. Extrem. Manuf. 2, 012004 (2020)) Pay Jun Liew, Ching Yee Yap, Jingsi Wang, Tianfeng Zhou and Jiwang Yan,, Surface modification and functionalization by electrical discharge coating: a comprehensive review on the functionalizing surface by coating should be referred.
We added this paper in the text as ref. no [67] in the list of references also.
